# Gaussian Mean Field Regularizes by Limiting Learned Information

**DOI:** 10.3390/e21080758

**Published:** 2019-08-03

**Authors:** Julius Kunze, Louis Kirsch, Hippolyt Ritter, David Barber

**Affiliations:** 1Computer Science, University College London, London WC1E 6BT, UK; 2The Swiss AI Lab (IDSIA), University of Lugano (USI) & University of Applied Sciences of Southern Switzerland (SUPSI), 6928 Manno, Switzerland; 3Alan Turing Institute, London NW1 2DB, UK

**Keywords:** information theory, variational inference, machine learning

## Abstract

Variational inference with a factorized Gaussian posterior estimate is a widely-used approach for learning parameters and hidden variables. Empirically, a regularizing effect can be observed that is poorly understood. In this work, we show how mean field inference improves generalization by limiting mutual information between learned parameters and the data through noise. We quantify a maximum capacity when the posterior variance is either fixed or learned and connect it to generalization error, even when the KL-divergence in the objective is scaled by a constant. Our experiments suggest that bounding information between parameters and data effectively regularizes neural networks on both supervised and unsupervised tasks.

## 1. Introduction

Bayesian machine learning is a popular framework for dealing with uncertainty in a principled way by integrating over model parameters rather than finding point estimates [1,2,3]. Unfortunately, exact inference is usually not feasible due to the intractable normalization constant of the posterior. A popular alternative is variational inference [4], where a tractable approximate distribution is optimized to resemble the true posterior as closely as possible. Due to its amenability to stochastic gradient descent [5,6,7,8], variational inference is scalable to large models and datasets.

The most common choice for the variational posterior is a factorized Gaussian. Outside of Bayesian inference, parameter noise has been found to be an effective regularizer [9,10,11], e.g., for training neural networks. In combination with L2-regularization, additive Gaussian parameter noise corresponds to variational inference with a Gaussian approximate posterior with fixed variance. Interestingly, it has been observed that flexible posteriors can perform worse than simple ones [12,13,14,15].

Variational inference follows the Minimum Description Length (MDL) principle [16,17,18], a formalization of Occam’s Razor. Loosely speaking, it states that of two models describing the data equally well, the “simpler” one should be preferred. However, MDL is only an objective for compressing the training data and the model and makes no formal statement about generalization to unseen data. Yet, generalization to new data is a key property of a machine learning algorithm.

Recent work [19,20,21,22] has proposed upper bounds on the generalization error as a function of the mutual information between model parameters and training data. It states that the gap between training and test error can be reduced by limiting the mutual information. However, to the best of our knowledge, these bounds and specific inference methods have so far not been linked.

In this work, we show that Gaussian mean field inference in models with Gaussian priors can be reinterpreted as point estimation in corresponding noisy models. This leads to an upper bound on the mutual information between model parameters and data through the data processing inequality. Our result holds for both supervised and unsupervised models. We discuss the connection to generalization bounds from Xu and Raginsky [19] and Bu et al. [20], suggesting that the Gaussian mean field aids generalization. In our experiments, we show that limiting model capacity via mutual information is an effective measure of regularization, further supporting our theoretical framework.

## 2. Regularization through the Mean Field

In our derivation, we denote a generic model as p(θ,D)=p(θ)p(D∣θ) with unobserved variables θ and data *D*. We refer to θ as the model parameters; however, in latent variable models, θ can also include the per-data point latent variables. The model consists of a prior p(θ) and a likelihood p(D∣θ). Ideally, one would like to find the posterior p(θ∣D)=p(D∣θ)p(θ)/Z, where Z=∫p(D∣θ)p(θ)dθ is the normalizer. However, calculating *Z* is typically intractable. Variational inference finds an approximation by maximizing the Evidence Lower Bound (ELBO)
(1)$$\begin{array}{c} \begin{array}{cc} \log p(D) & \geq \log p\left(D\right)-D_{\mathrm{KL}}\left(q(\theta )\|p(\theta |D)\right)\\ & =E_{q(\theta )}\log p\left(D|\theta \right)-D_{\mathrm{KL}}\left(q(\theta )\|p(\theta )\right) \end{array} \end{array}$$
w.r.t. the approximate posterior q(θ). Our focus in this work lies on Gaussian mean field inference, so *q* is a fully-factorized normal distribution with a learnable mean μ and variance $\sigma^{2}$. The prior is also chosen to be component-wise independent $p\left(\theta \right)=N\left(0,\sigma_{p}^{2}I\right)$. The generative and inference models for this setting are shown in Figure 1a.

### 2.1. Fixed-Variance Gaussian Mean Field Inference

When the variance $\sigma^{2}$ of the approximate posterior is fixed, the ELBO can be written as:(2)$$\begin{array}{c} E_{\theta ∼N\left(\mu ,\sigma^{2}I\right)}\log p\left(D|\theta \right)-\frac{1}{2\sigma_{p}^{2}}\sum_{i}\mu_{i}^{2}+c \end{array}$$
which is optimized with respect to μ. We use i∈{1,…,K} to denote the parameter index and *c* for constant terms.

To show how the Gaussian mean field implicitly limits learned information, we extend the model with a noisy version of the parameters $\tilde{\theta }∼p\left(\tilde{\theta }|\theta \right)$ and let the likelihood depend on those noisy parameters. We choose the noise distribution to be the same as the inference distribution for the original model and find a lower bound on the log-joint of the noisy model. This leads to the same objective as mean field variational inference in the original model.

Specifically, we define the noisy model $p^{\prime }\left(\theta ,\tilde{\theta },D\right)=p^{\prime }\left(\theta \right)p^{\prime }\left(\tilde{\theta }|\theta \right)p^{\prime }\left(D|\tilde{\theta }\right)$ as visualized in Figure 1b. We use $p^{\prime }$ to emphasize the distinction between distributions of the modified noisy model and the original one. As in the original model, $\theta ∼N\left(0,\sigma_{p}^{2}I\right)$ represents the parameters (with the same prior), i.e., $p\left(\theta \right)=p^{\prime }\left(\theta \right)$. We denote the noisy parameters as $\tilde{\theta }∼N\left(\theta ,\sigma^{2}\right)$. The likelihood remains unchanged, i.e., $p^{\prime }\left(D|\tilde{\theta }\right)=p\left(D|\theta \right)$, except that it now depends on the noisy parameters instead of the “clean” ones.

We now show that maximizing a lower bound on the log-joint probability of the noisy model results in an identical objective as for variational inference in the clean model
(3)$$\begin{array}{c} \log p^{\prime }\left(D,\theta \right) \end{array}$$
(4)$$\begin{array}{cc} = & \log\int p^{\prime }\left(D|\tilde{\theta }\right)p^{\prime }\left(\tilde{\theta }|\theta \right)\;d\tilde{\theta }+\log p^{\prime }\left(\theta \right) \end{array}$$
(5)$$\begin{array}{cc} \geq & E_{\tilde{\theta }∼N\left(\theta ,\sigma^{2}I\right)}\log p^{\prime }\left(D|\tilde{\theta }\right)-\frac{1}{2\sigma_{p}^{2}}\sum_{i}\theta_{i}^{2}+c \end{array}$$
(6)$$\begin{array}{cc} = & E_{\tilde{\theta }∼N\left(\mu ,\sigma^{2}I\right)}\log p^{\prime }\left(D|\tilde{\theta }\right)-\frac{1}{2\sigma_{p}^{2}}\sum_{i}\mu_{i}^{2}+c \end{array}$$
where Equation (Equation 5) follows from Jensen’s inequality as in Equation (Equation 1). In the final equation, we have replaced θ with μ (which is simply a change of names, since we are maximizing the objective over this free variable) to emphasize that the objective functions are identical.

Since *D* is independent of θ given $\tilde{\theta }$, the joint $p(\theta ,\tilde{\theta },D)$ forms a Markov chain, and the data processing inequality [23] limits the mutual information I(D,θ) between learned parameters and data through:(7)$$\begin{array}{c} I\left(D,\theta \right)\leq I\left(\tilde{\theta },\theta \right) \end{array}$$

The upper bound is given by:(8)$$\begin{array}{cc} I(\tilde{\theta },\theta ) & =H\left(\tilde{\theta }\right)-H\left(\tilde{\theta }|\theta \right)=\frac{K}{2}\log\left(1+\frac{\sigma_{p}^{2}}{\sigma^{2}}\right) \end{array}$$
where *K* denotes the number of parameters. Here, we exploit that θ and $\tilde{\theta }|\theta$ are Gaussian with $H\left(\tilde{\theta }\right)=\frac{K}{2}\log2\pi e\left(\sigma^{2}+\sigma_{p}^{2}\right)$ and $H\left(\tilde{\theta }|\theta \right)=\frac{K}{2}\log2\pi e\sigma^{2}$. This quantity is known as the capacity of channels with Gaussian noise in signal processing [23]. Intuitively, a high prior variance $\sigma_{p}^{2}$ corresponds to a large capacity, while a high noise variance $\sigma^{2}$ reduces it. Any desired capacity can be achieved by simply adjusting the signal-to-noise ratio $\sigma_{p}^{2}/\sigma^{2}$.

### 2.2. Generalization Error vs. Limited Information

Intuitively, we characterize overfitting as learning too much information about the training data, suggesting that limiting the amount of information extracted from the training data into the hypothesis should improve generalization. This idea was recently formalized by Xu and Raginsky [19], Bu et al. [20], Bassily et al. [21], Russo and Zou [22], showing that limiting mutual information between data and learned parameters bounds the expected generalization error under certain assumptions.

Specifically, their work characterizes the following process: Assume that our training dataset is sampled from a true distribution $p_{t}\left(D\right)$. Based on this training set, a learning algorithm subsequently returns a distribution over hypotheses given by $p_{t}\left(\theta |D\right)$. The process defines mutual information $I_{t}\left(D,\theta \right)$ on the joint distribution $p_{t}\left(D,\theta \right)=p_{t}\left(D\right)p_{t}\left(\theta |D\right)$. Under certain assumptions on the loss function, Xu and Raginsky [19] derived a bound on the generalization error of the learning algorithm in expectation over this sampling process. Bu et al. [20] relaxed the condition on the loss and proved the applicability to a simple estimation algorithm involving L2-loss.

Exact Bayesian inference returns the true posterior p(θ∣D) on a model p(θ,D). The theorem then states that a bound on I(D,θ) limits the expected generalization error as described in Bu et al. [20] if the model captures the nature of the generating process in the marginal p(D)=∫dθp(θ)p(D∣θ). This is a common assumption necessary to justify any (variational) Bayesian approach.

Exact inference is intractable on deep models, and instead, one typically learns variational or point estimates for the posterior. That is also true for the objective on the noisy model above, where we only used a point estimate as given by Equation (Equation 6). Therefore, the assumption of exact inference is not met. Yet, we believe that those bounds motivate the expectation that variational inference aids generalization by limiting the learned information. If we performed exact inference on the noisy model in the last section, the given mutual information would imply a bound on generalization error as implied by Xu and Raginsky [19] and Bu et al. [20]. Therefore, we are optimistic that the gap between variational inference and those generalization bounds can be closed either by performing more accurate inference in the noisy model or by taking the dynamics of the training algorithm into account when bounding mutual information (see Section 5.2 for further discussion).

### 2.3. Learned-Variance Gaussian Mean Field Inference

The variance in Gaussian mean field inference is typically learned for each parameter [8,24,25]. Similar to when the variance in the approximate posterior is fixed, one can obtain a capacity constraint. This is the case even for a generalization of the objective from Equation (Equation 1) where the divergence term $D_{\mathrm{KL}}\left(q(\theta )\|p(\theta )\right)$ is scaled by some factor β>0. Higgins et al. [26] proposed using β>1 to learn “disentangled” representations in variational autoencoders. Further, β is commonly annealed from 0–1 for expressive models (e.g., Blundell et al. [25], Bowman et al. [27], Sønderby et al. [28].) In the following, we quantify a general capacity depending on β, where β=1 recovers the standard variational objective. For notational simplicity, we here assume a prior variance of $\sigma_{p}^{2}=1$. It is straight-forward to adapt the derivation to the general case.

In this case, the objective can be written as: (9)$$\begin{array}{c} \begin{array}{c} E_{\theta ∼N\left(\mu ,\sigma^{2}\right)}\log p\left(D|\theta \right)+\frac{\beta }{2}\sum_{i}\left(\log\sigma_{i}^{2}-\sigma_{i}^{2}-\mu_{i}^{2}-1\right) \end{array} \end{array}$$
where now, both μ and $\sigma^{2}$ represent learned vectors and $N\left(\mu ,\sigma^{2}\right)$ denotes a variable composed of pairwise independent Gaussian components with means and variances given by the elements of μ and $\sigma^{2}$.

Similar to the previous section, we show a lower bound on the log-joint of a new noisy model to be identical to Equation (Equation 9). Specifically, we define the noisy model $p^{\prime }\left(\theta ,\sigma^{2},\tilde{\theta },D\right)=p^{\prime }\left(\theta \right)p^{\prime }\left(\sigma^{2}\right)p^{\prime }\left(\tilde{\theta }|\theta ,\sigma^{2}\right)p^{\prime }\left(D|\tilde{\theta }\right)$ (Figure 1c), with independent priors $\theta_{i}∼N\left(0,\frac{1}{\beta }\right)$ and $\sigma_{i}^{2}∼\Gamma \left(\frac{\beta }{2}+1,\frac{\beta }{2}\right)$, where Γ(·,·) denotes the Gamma distribution. As previously done Section 2.1, we define the noise-injected parameters as $\tilde{\theta }∼N\left(\theta ,\sigma^{2}\right)$ and likelihood as $p^{\prime }\left(D|\tilde{\theta }\right)=p\left(D|\theta \right)$.

The priors are chosen so that with Jensen’s inequality, we find a lower bound on the log-joint probability of this model that recovers the objective from Equation (Equation 9): (10)$$\begin{array}{c} \begin{array}{cc} & \log p^{\prime }\left(D,\theta ,\sigma^{2}\right)\\ = & \log\int p^{\prime }\left(D|\tilde{\theta }\right)p^{\prime }\left(\tilde{\theta }|\theta ,\sigma^{2}\right)\;d\tilde{\theta }+\log p^{\prime }\left(\theta \right)+\log p^{\prime }\left(\sigma^{2}\right)\\ \geq & E_{\tilde{\theta }∼N\left(\theta ,\sigma^{2}\right)}\log p^{\prime }\left(D|\tilde{\theta }\right)+\sum_{i}\left(\log p^{\prime }\left(\theta_{i}\right)+\log p^{\prime }\left(\sigma_{i}^{2}\right)\right)\\ = & E_{\tilde{\theta }∼N\left(\mu ,\sigma^{2}\right)}\log p^{\prime }\left(D|\tilde{\theta }\right)+\frac{\beta }{2}\sum_{i}\left(\log\sigma_{i}^{2}-\sigma_{i}^{2}-\mu_{i}^{2}\right)+c \end{array} \end{array}$$

In the noisy model, the data processing inequality and the independence of dimensions implies a bound:(11)$$\begin{array}{c} I\left(D,\left(\theta ,\sigma^{2}\right)\right)\leq I\left(\tilde{\theta },\left(\theta ,\sigma^{2}\right)\right)=\sum_{i}I\left({\tilde{\theta }}_{i},\left(\theta_{i},\sigma_{i}^{2}\right)\right) \end{array}$$
where the capacity $I({\tilde{\theta }}_{i},\left(\theta_{i},\sigma_{i}^{2}\right))$ per dimension is derived in Appendix A.

Figure 2 shows numerical results for various values of β. Standard variational inference (β=1) results in a capacity of 0.45bits per dimension. We observe that higher β corresponds to smaller capacity, which is given by the mutual information $I({\tilde{\theta }}_{i},\left(\theta_{i},\sigma_{i}^{2}\right))$ between our new latent $\left(\theta_{i},\sigma_{i}^{2}\right)$ and ${\tilde{\theta }}_{i}$. This formalizes the intuition that a higher weight of the complexity term in our objective increases regularization by decreasing a limit on the capacity.

### 2.4. Supervised and Unsupervised Learning

The above derivations apply to any learning algorithm that is purely trained with Gaussian mean field inference. This covers supervised and unsupervised tasks.

In supervised learning, the training data typically consist of pairs of inputs and labels, and a loss is assigned to each pair that depends on the trained model, e.g., neural network parameters. When all parameters are learned with one of the discussed mean field methods, the given bounds apply.

The derivation also comprises unsupervised methods with per-data latent variables and even amortized inference such as variational autoencoders [8,24], again as long as all learned variables are learned via Gaussian mean field inference. While this might be helpful to find generalizing representations, the focus of the experiments is on validating the generalizing behaviour of the Bayesian mean field variational approach on neural network parameters for overfitting regimes, namely a small dataset and complex models.

### 2.5. Flexible Variational Distributions

The objective function for variational inference is maximized when the approximate posterior is equal to the true one. This motivates the development of flexible families of posterior distributions [8,29,30,31,32,33,34,35,36]. In the case of exact inference, a bound on generalization as discussed in Section 2.2 only applies if the model itself has finite mutual information between data and parameters. However, estimating mutual information is generally a hard problem, particularly in high-dimensional, non-linear models. This makes it hard to state a generic bound, which is why we focus on the case of Gaussian mean field inference.

## 3. Related Work

### 3.1. Regularization in Neural Networks

The Gaussian mean field is intimately related to other popular regularization approaches in deep learning: As is apparent from Equation (Equation 6), the fixed-variance Gaussian mean field applied to training neural network weights is equivalent to L2-regularization (weight decay) combined with Gaussian parameter noise [9,10,11] on all network weights. Molchanov et al. [37] showed that additive parameter noise results in multiplicative noise on the unit activations. The resulting dependencies between noise components on the layer output can be ignored without significantly changing empirical results [38]. This is in turn equivalent to scaled Gaussian dropout [24].

### 3.2. Information Bottlenecks

The information bottleneck principle by Tishby et al. [39], Shamir et al. [40] aims to find a representation *Z* of some input *X* that is most useful to predict an output *Y*. For this purpose, the objective is to maximize the amount of information I(Y,Z) the representation contains about the output under a bounded amount of information I(X,Z) about the input:(12)$$\begin{array}{c} \max_{I(X,Z)<C}I\left(Y,Z\right) \end{array}$$
They described a training procedure using the softly-constrained objective:(13)$$\begin{array}{c} \min L_{IB}=\min I\left(X,Z\right)-\beta I\left(Y,Z\right) \end{array}$$
where β>0 controls the trade-off.

Alemi et al. [41] suggested a variational approximation for this objective. For the task of reconstruction, where labels *Y* are identical to inputs *X*, this results exactly in the β-VAE objective [42,43]. This is in accordance with our result from Section 2.3 that there is a maximum capacity per latent dimension that decreases for higher β. Setting β>1, as suggested by Higgins et al. [26], for obtaining disentangled representations, corresponds to lower capacity per latent component than achieved by standard variational inference.

Both Tishby et al. [39] and Higgins et al. [26] introduced β as a trade-off parameter without a quantitative interpretation. With our information-theoretic perspective, we quantify the implied capacity and provide a link to the generalization error. Further, both methods are concerned with the information in the latent representation. They do not limit the mutual information with the model parameters, leaving them vulnerable to model overfitting under our theoretical assumptions. We experimentally validated this vulnerability and explore the effect of filling this gap by applying Gaussian mean field inference to the model parameters.

### 3.3. Information Estimation with Neural Networks

Multiple recent techniques [44,45,46] proposed the use of neural networks for obtaining a lower bound on the mutual information. This is useful in settings when we want to maximize mutual information, e.g., between the data and a lower-dimensional representation. In contrast, we show that Gaussian variational inference on variables with a Gaussian prior implicitly places an upper bound on the mutual information between these variables and the data and explore its regularizing effect.

## 4. Experiments

In this section, we analyse the implications of applying Gaussian mean field inference of a fixed scale to the model parameters in the supervised and unsupervised context. Our theoretical results suggest that varying the capacity will affect the generalization capability, and we show this effect on small data regimes and how it changes with the training set size. Furthermore, we investigate whether capacity is the only predictor for generalization or whether varying priors and architectures also have an effect. Finally, we demonstrate qualitatively how the capacity bounds are reflected in Fashion MNIST reconstruction.

### 4.1. Supervised Learning

We begin with a supervised classification task on the CIFAR10 dataset, training only on a subset of the first 5000 samples. We used 63 × 3 convolutional layers with 128 channels each followed by a ReLU activation function, every second of which implemented striding of 2 to reduce the input dimensionality. Finally, the last layer was a linear projection, which parameterized a categorical distribution. The capacity of each parameter in this network was set to specific values given by Equation (Equation 8).

Figure 3 shows that decreasing the model capacity per dimension (by increasing the noise) reduced the training log-likelihood and increased the test log-likelihood until both of them meet at an optimal capacity. One can observe that very small capacities led to a signal that was too noisy, and good predictions were no longer possible. In short, regimes of underfitting and overfitting were generated depending on the capacity.

### 4.2. Unsupervised Learning

We now evaluate the regularizing effect of fixed-scale Gaussian mean field inference in an unsupervised setting for image reconstruction on the MNIST dataset. Therefore, we used a VAE [6] with 2 latent dimensions and a 3-layer neural network parameterizing the conditional factorized Gaussian distribution. As usual, it was trained using the free energy objective, but different from the original work, we also used Gaussian mean field inference for the model parameters. Again, we used a small training set of 200 examples for the following experiments if not denoted otherwise.

#### 4.2.1. Varying Model Capacity and Priors

In our first experiment, we analysed generalization by inspecting the test evidence lower bound (ELBO) when varying the model capacity, which can be seen in Figure 4a. Similar to the supervised case, we can observe that there was a certain model capacity range that explained the data very well, while less or more capacity resulted in noise drowning and overfitting, respectively.In the same figure, we also investigated whether the information-theoretic model capacity can predict generalization independently of the specific prior distribution. Since we merely state an upper bound on mutual information in Section 2.1, the prior may have an effect in practice, which cannot be explained only by the capacity. Figure 4a shows that indeed, while the general behaviour remained the same for different model priors, the generalization error was not entirely independent. Furthermore, the observation that all curves descended with larger capacities, for all priors, suggests that weight decay [47] of fixed scale without parameters noise was not sufficient to regularize arbitrarily large networks. In Figure 4b, we investigated the extreme case of dropping the prior entirely and switching to maximum-likelihood learning instead by using an improper uniform prior. This approach recovered Gaussian dropout [24,48]. Dropping the prior set the bottleneck capacity to infinity and should lead to worse generalization. Comparing the test ELBO of this Gaussian dropout variant to the original Gaussian mean field inference in Figure 4b confirmed this result for larger capacities. For larger noise scales, generalization was still working well, a result that was not explained in our information-theoretic framework, but plausible due to the deployed limited architecture.

#### 4.2.2. Varying Training Set Size

Figure 5a shows how limiting the capacity affects the test ELBO for varying amounts of training data. Models with very small capacity extracted less information from the data into the model, thus yielding a good test ELBO somewhat independent of the dataset size. This is visible as a graph that ascends very little with more training data (e.g., total model capacity of 330 kbits). Note that we here report the capacity of the entire model, which is the sum of the capacities for each parameter. In order to improve the test ELBO, more information from the data had to be extracted into the model. However, clearly, this led to non-generalizing information being extracted when the dataset was small, leading to overfitting. Only for larger datasets, the extracted information generalized. This is visible as a strongly-ascending test ELBO with larger dataset sizes and bad generalization for small datasets. We can therefore conclude that the information bottleneck needs to be chosen based on the amount of data that are available. Intuitively, when more information is available, the more information should be extracted into the model.

#### 4.2.3. Varying Model Size

Furthermore, we inspected how the size of the model (here, in terms of the number of layers) affected generalization in Figure 5b. Similar to varying the prior distribution, we were interested in how well the total capacity predicted generalization and the role the architecture plays. It can be observed that larger networks were more resilient to larger total capacities before they started overfitting. This indicates that the total capacity was less important than the individual capacity (i.e., noise) per parameter. Nevertheless, larger networks were more prone to overfitting for very large model capacities. This makes sense as their functional form was less constrained, an aspect that was not captured by our theory.

#### 4.2.4. Qualitative Reconstruction

Finally, we plot test reconstruction means for the binarized Fashion MNIST dataset under the same setup for various capacities in Figure 6. In accordance with the previous experiments, we observed that if the capacity was chosen too small, the model was not learning anything useful, while too large capacities resulted in overconfidence. This can be observed in most means being close to either 0–1. An intermediate capacity, on the other hand, made sensible predictions (given that it was trained only on 200 samples) with sensible uncertainty, visible through grey pixels that correspond to high entropy.

## 5. Discussion

In this section, we discuss how the capacity can be set, as well as the effect of the model architecture and learning dynamics.

### 5.1. Choosing the Capacity

We have obtained a new trade-off parameter, the capacity, that has a simple quantitative interpretation: It determines how many bits to extract maximally from the training set. In contrast, for the β parameter introduced in Tishby et al. [39] and Higgins et al. [26], a clear interpretation is not known. Yet, it may still be hard to set the capacity optimally. Simple mechanisms such as evaluation of a validation set to determine its value may be used. We expect that more theoretically-rigorous methods could be developed.

Furthermore, in this paper, we focused on the regularization that Gaussian mean field inference implies on the model parameters. The same concept is valid for data-dependent latent variables, for instance in VAEs, as discussed in Section 2.4. In VAEs, Gaussian mean field inference on the latent variables leads to a restricted latent capacity, but leaves the capacity of the model unbounded. This leaves VAEs vulnerable to model overfitting, as demonstrated in the experiments, and setting β as done in Higgins et al. [26] is not sufficient to control complexity. This motivates the limitation of capacity between the data and both per-data point latent variables and model parameters. The interaction between the two is an interesting future research direction.

### 5.2. Role of Learning Dynamics and Architecture

As discussed in Section 2.2, it is necessary to perform exact inference in the noisy model for the bounds on the generalization error to hold. However, this assumption was not met. In practice, $p_{t}\left(\theta |D\right)$ encodes the complete learning algorithm, which in deep learning typically includes parameter initialization and the dynamics of the stochastic gradient descent optimization.

Our experiments confirmed the relevance of the aforementioned factors: L2-regularization works in practice, even though no noise was added to the parameters. This could be explained by the fact that noise was already implicitly added through stochastic gradient descent [49] or through the output distribution of the network. Similarly, Gaussian dropout [9,10,11] without a prior on the parameters helped generalization. Again, early stopping combined with a finite reach of gradient descent steps effectively shaped a prior of finite variance in the parameter space. This could also formalize why the annealing schedule employed by Blundell et al. [25], Bowman et al. [27] and Sønderby et al. [28] was effective.

Since these other factors affect generalization, quantifying mutual information $I_{t}\left(\theta ,D\right)$ of the actual distribution due to the learning dynamics might be a promising approach to explain why neural networks often generalize well on their own. This idea is in accordance with recent work that links the learning dynamics of small neural networks to generalization behaviour [50].

On the other hand, the architecture choice also had an influence on generalization. This does not contradict our theory, we since we only formulated an upper bound on mutual information. Tightening this bound based on the model architecture and output distribution is usually hard, as discussed in Section 2.5, but might be possible.

Another promising direction would be to sample approximately from the exact posterior on network parameters (i.e., as done by Marceau-Caron and Ollivier [51]), on a capacity-limited architecture, instead of the usual approach of point estimation. In the limit of infinite training time, this would fully realize the discussed bound on the expected generalization error.

## 6. Conclusions

We explored an information-theoretic perspective on the regularizing effects observed in Gaussian mean field approaches. The derivation featured a capacity that can be naturally interpreted as a limit on the amount of information extracted about the given data by the inferred model. We validated its practicality for both supervised and unsupervised learning.

How this capacity should be set for parameters and latent variables depending on the task and data is an interesting direction of research. We exploited a theoretical link of mutual information and generalization error. While this work is restricted to the Gaussian mean field, incorporating the effect of learning dynamics on mutual information in the future work might allow understanding why overparameterized neural networks still generalize well to unseen data.

## Figures and Tables

**Figure 1 entropy-21-00758-f001:**
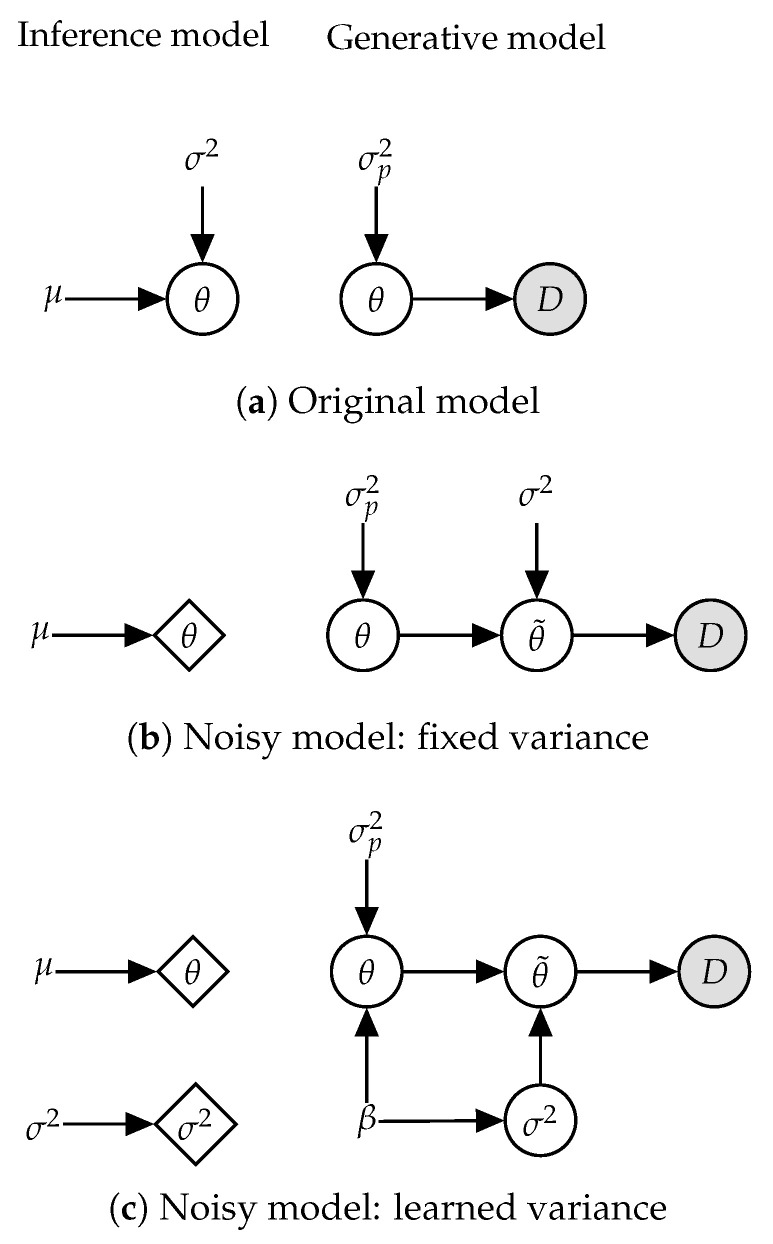
Gaussian mean field inference on model parameters θ with a Gaussian prior (**a**) can be reinterpreted as optimizing a point estimate on a model with injected noise, both when variance is fixed (**b**) and learned (**c**). For the latter case, we show this for the more general case where the complexity term in the objective is scaled by a constant β>0, with β=1 recovering variational inference.

**Figure 2 entropy-21-00758-f002:**
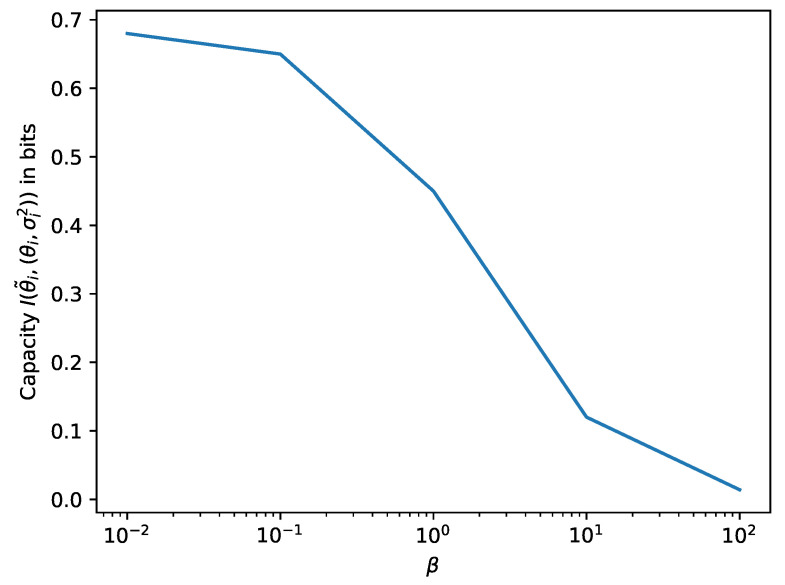
Relationship between β and capacity $I({\tilde{\theta }}_{i},\left(\theta_{i},\sigma_{i}^{2}\right))$ per parameter dimension in Gaussian mean field inference with learned variance and the complexity term scaled by β>0.

**Figure 3 entropy-21-00758-f003:**
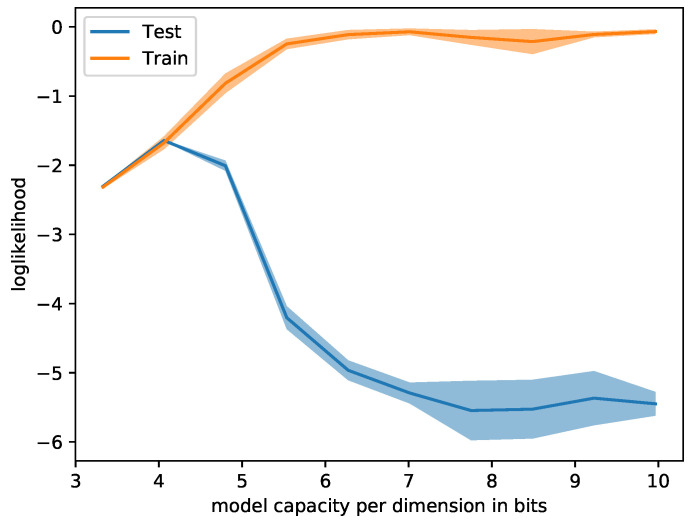
Classifying CIFAR10 with varying model capacities. Large capacities lead to overfitting, while small capacities drown the signal in noise. Each configuration was evaluated 5 times; the mean and standard deviation are displayed.

**Figure 4 entropy-21-00758-f004:**
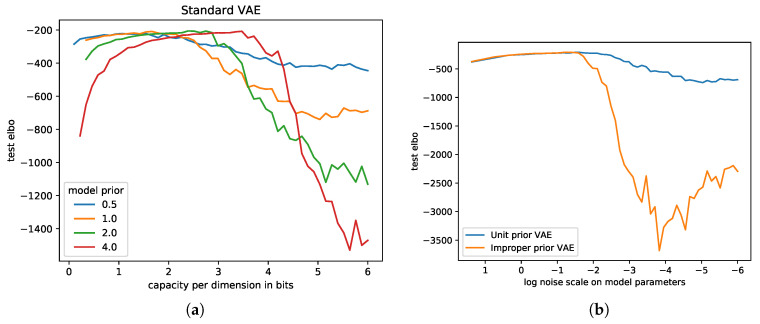
MNIST test reconstruction with a variational autoencoder training on 200 samples for various capacities and Gaussian priors. (**a**) The test Evidence Lower Bound (ELBO) is not invariant when varying the prior on the model parameters. Nevertheless, the first increasing and then decreasing trend when changing the capacity remains; (**b**) Using an improper prior, similar to just using Gaussian dropout on the weights, leads to an accelerated decrease of generalization for smaller noise scales.

**Figure 5 entropy-21-00758-f005:**
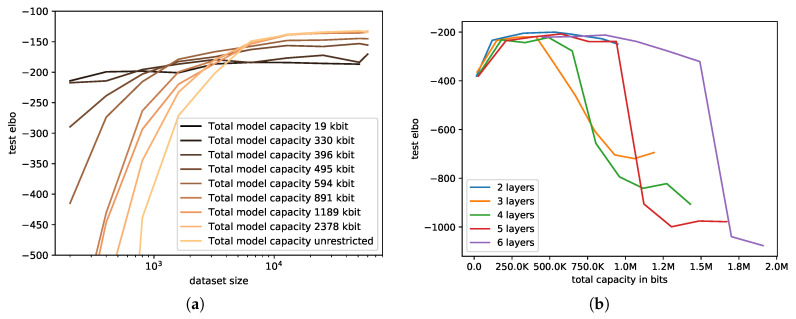
MNIST test reconstruction with a VAE; training on varying dataset sizes, architectures, and model capacities. (**a**) Varying the number of samples. Depending on the size of the dataset, higher capacities of the model are required to fit all the data points; (**b**) Varying architecture. Overfitting is not getting worse for more layers if the capacity is low enough. More layers do overfit only for higher capacities.

**Figure 6 entropy-21-00758-f006:**
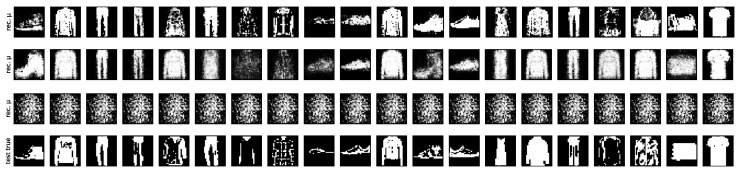
Test reconstruction means for binarized Fashion MNIST trained on 200 samples with per-parameter capacities of 5, 2, and 1bits (**top**) compared to the true data (**bottom**).

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
