# Peer review of "Gaussian Mean Field Regularizes by Limiting Learned Information"

_entropy, 2019, doi:10.3390/e21080758_

Round 1

Reviewer 1 Report

This paper explains the regularizing effects observed in Gaussian mean field approaches from an information-theoretic perspective. The overall presentation is clear and the topic seems pretty interesting. However, the scientific significance of this work is not adequate. I, thus, suggest to decline the publication.

I agree the factorized Gaussian posterior is used in some work. But this assumption itself introduces model error in most applications. From an applied point of view, adopting this method is totally fine. However, in the content of quantifying the mutual information and other types of information, should a more careful analysis by taking into account both the model error and useful information be implemented? There is no discussion about the potential model error here.

I also suggest to perform some simple experiments, in particular some perfect model experiments, in order to explain with more details the mutual information and the error.

Reviewer 2 Report

Please see attached file for specific comments.

Round 2

Reviewer 1 Report

I don't think the authors addressed my previous comments, which admittedly are not easy to address. But given the current self-contained description and study of the content, I think the paper worth publication. 

Reviewer 2 Report

My concerns have been addressed and I recommend the manuscript for publication.